# Interpretable Uncertainty-Aware Deep Regression with Cohort Saliency Analysis for Three-Slice CT Imaging Studies

**Nouman Ahmad**[1]                                             NOUMAN.AHMAD@UU.SE
**Johan Öfverstedt**[1]                                         JOHAN.OFVERSTEDT@UU.SE
**Sambit Tarai**[1,2]                                           SAMBIT.TARAI@UU.SE
**Göran Bergström**[3,4]                                        GORAN.BERGSTROM@HJL.GU.SE
**Håkan Ahlström**[1,2]                                         HAKAN.AHLSTROM@UU.SE
**Joel Kullberg**[1,2]                                          JOEL.KULLBERG@UU.SE

[1] *Radiology, Department of Surgical Sciences, Uppsala University, Uppsala, Sweden*

[2] *Antaros Medical, Mölndal, Sweden*

[3] *Department of Molecular and Clinical Medicine, Institute of Medicine, Sahlgrenska Academy, University of Gothenburg, Gothenburg, Sweden*

[4] *Department of Clinical Physiology, Sahlgrenska University Hospital, Region Västra Götaland, Gothenburg, Sweden*

**Editors:** Accepted for publication at MIDL 2024

## Abstract

Obesity is associated with an increased risk of morbidity and mortality. Achieving a healthy body composition, which involves maintaining a balance between fat and muscle mass, is important for metabolic health and preventing chronic diseases. Computed tomography (CT) imaging offers detailed insights into the body's internal structure, aiding in understanding body composition and its related factors. In this feasibility study, we utilized CT image data from 2,724 subjects from the large metabolic health cohort studies SCAPIS and IGT. We train and evaluate an uncertainty-aware deep regression based ResNet-50 network, which outputs its prediction as mean and variance, for quantification of cross-sectional areas of liver, visceral adipose tissue (VAT), and thigh muscle. This was done using collages of three single-slice CT images from the liver, abdomen, and thigh regions. The model demonstrated promising results with the evaluation metrics – including R-squared ($R^2$) and mean absolute error (MAE) for predictions. Additionally, for interpretability, the model was evaluated with saliency analysis based on Grad-CAM (Gradient-weighted Class Activation Mapping) at stages 2, 3, and 4 of the network. Deformable image registration to a template subject further enabled cohort saliency analysis that provide group-wise visualization of image regions of importance for associations to biomarkers of interest. We found that the networks focus on relevant regions for each target, according to prior knowledge. The source code is available at: https://github.com/noumannahmad/dr_3slice_ct.

**Keywords:** Body composition, Computed tomography, Deep regression

## 1. Introduction

Understanding metabolic health is important for understanding the risks and mechanisms of type 2 diabetes (T2D) and cardiovascular disease (CVD) (Stefan and Schulze, 2023). Body composition is associated with metabolic health. The analysis of body composition involves

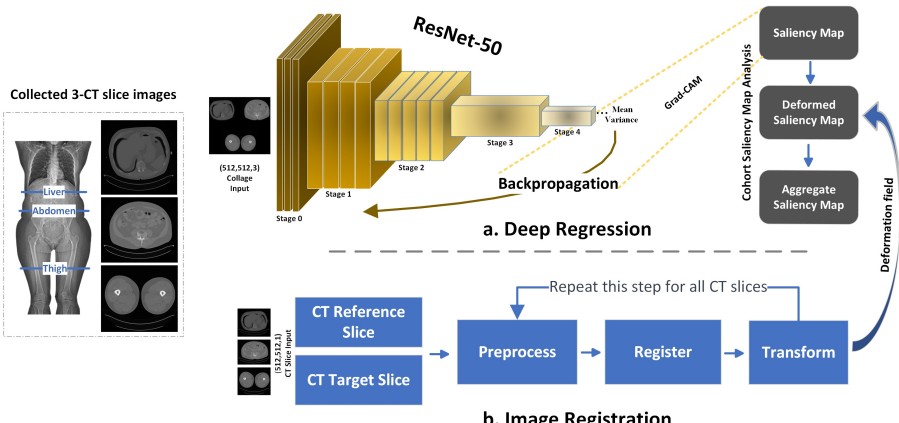

Figure 1: Illustration of the (collected) three-slice CT image data and the image analysis pipelines used: (a) Deep regression using a ResNet-50 network for prediction of clinical variables of interest, and generation of the cohort saliency map through aggregation of individual saliency maps generated with Grad-CAM, and (b) the registration pipeline used to transform the saliency maps into a common space.

examining the quantities and distribution of both fatty and non-fatty tissues, including adipose tissue, muscle, liver, and bone. Precise measurement of body composition is crucial for understanding the mechanisms and progression of cardiometabolic diseases, particularly T2D and CVD, thereby aiding in their prediction and prevention (Sneed and Morrison, 2021). Non-alcoholic fatty liver disease (NAFLD) (Gaggini et al., 2013; Ariya et al., 2021), fat tissues and muscles, especially visceral adipose tissue (VAT), the fat stored around abdominal organs are associated with metabolic risk (Janochova et al., 2019; Vasamsetti et al., 2023). NAFLD and VAT can cause insulin resistance, a key factor in the development of T2D and metabolic disturbances, increasing the risk of T2D, coronary artery disease, and CVD (Samaras et al., 2010).

Computed tomography (CT) is an imaging technique that allows detailed analysis of body composition (Kullberg et al., 2017; Ahmad et al., 2023). SCAPIS and IGT, are two large Swedish cohort studies that include CT imaging for detailed studies of body composition and its link to cardiometabolic disease and Chronic Obstructive Pulmonary Disease (Bergström et al., 2015). The IGT study focuses on investigating the influence of gut microbiota on glucose dysregulation and cardiometabolic risk (Molnar et al., 2023).

In this feasibility study, we propose to use the ResNet-50 network, as illustrated in Figure 1 for the quantification of variables such as liver, VAT, and thigh muscle areas from a collage of three single-slice CT images, as applied to the SCAPIS and IGT cohort studies. SCAPIS and IGT include three single-slice CT imaging data for body composition analysis. The acquisition of only a few slices minimizes ionizing radiation (mean 0.2 mSv) (Bergström et al., 2015) compared to volumetric CT scans while still providing body composition information. For uncertainty awareness, the network predicts both mean and variance, providing an estimate of its ability to predict the target value. For interpretability, Grad-CAM is used

at different stages of the network to generate saliency maps highlighting important regions for each prediction, and deformable image registration (on the CT images) is employed to align the resulting saliency maps into a common template space, enabling a cohort saliency map to identify shared important regions across the entire cohort. The network exhibits promising results in predicting the selected targets according to multiple evaluation measures, suggesting the viability of deep regression to predict clinical parameters from images acquired using the three single-slice CT imaging protocol. The saliency maps highlight areas expected to be of high importance to the prediction targets, reinforcing trust in both the cohort saliency analysis methodology and the proposed deep regression method.

## 2. Background and related work

Deep learning techniques (Elhakim et al., 2023) are increasingly favored due to their proficiency in automatic feature extraction and decision-making. These advanced methods are effective in segmenting and measuring the liver, different types of fat, thigh muscles, and other tissues and organs. Beyond segmentation, these techniques can be used for classification and aid in the estimation and prediction of body composition parameters, age, BMI, and other non-imaging parameters and biomarkers, as described in studies including MR images from the UK Biobank Study (Langner et al., 2020, 2022; Starck et al., 2023). From CT images of the thorax and abdomen, a ResNet-18 network has previously been used for age prediction (Kerber et al., 2023).

Numerous deep learning network architectures have been developed for a variety of tasks, with ResNet-50 (He et al., 2016) being widely utilized due to its skip connections facilitating efficient and effective training. To address the challenge of limited data in medical imaging, transfer learning methods are a powerful family of techniques for leveraging large amounts of training data in one domain to solve tasks in a different domain where data is limited. Since deep learning models typically require large amounts of data for effective training, using transfer learning (Kim et al., 2022) to retrain models with medical images can substantially enhance performance, proving particularly beneficial in this context.

The integration of deep learning in medical imaging decision-making poses challenges related to trustworthiness due to the black-box nature of deep learning systems, but these can be partially mitigated by enhancing the interpretability and uncertainty awareness of these systems. For network interpretability, Grad-CAM (Gradient-weighted Class Activation Mapping) (Selvaraju et al., 2016) is one of the commonly used methods in deep learning that highlights features and spatial regions of importance for a network's prediction by computing the gradient of each feature w.r.t. a selected target class/neuron.

Image registration is a technique in medical imaging that aligns (multiple) images to a common reference space (Fox et al., 2008). This standardization enables comparative studies, such as Imiomics analysis (Strand et al., 2017) and the generation of cohort saliency maps (Langner et al., 2020). In (Ekström et al., 2020), a graph-cut-based method for image registration was proposed. Building on this methodology, in (Jönsson et al., 2022), a registration technique for PET-CT images was proposed, incorporating anatomical and tissue-wise masks to enhance the robustness and accuracy of the registration.

## 3. Methodology

### 3.1. Dataset overview and ethical statement

This study used data from two large cohort studies, SCAPIS and IGT. The primary objective of the SCAPIS study is to explore T2D, CVD, and COPD (Bergström et al., 2015). SCAPIS included 30,154 participants, comprising both males and females aged between 50 and 64 years who volunteered to participate. The SCAPIS study data was collected in six university hospitals across Sweden in the time period from 2013 to 2018. For this study, a randomly selected subset of data from participants recruited in Gothenburg was used.

The IGT (Molnar et al., 2023) study is a parallel cohort to SCAPIS, with a focus on individuals at high risk of developing T2D. This study particularly investigates the role of the gut microbiota in the dysregulation of glucose and the consequent development of CVD. The IGT cohort comprised 1,965 participants with varying forms of glucose dysregulation. Both SCAPIS and IGT cohorts were examined using a common CT imaging protocol.

Our study received approval from the Swedish Ethical Review Authority (Dnr 2021-05856-01), and all participants provided their informed consent in written form.

All CT images were acquired according to a predefined protocol, featuring an image dimension of $512 \times 512 \times 1$, a field of view of 500mm, and a slice thickness of 5mm. For the analysis, three specific CT slices were utilized, which we refer to as: *liver*, *abdomen*, and *thigh*. The liver slices were reconstructed from lung scans. The abdomen slice was positioned above the crista edge and centrally aligned with the L4 vertebra. The thigh slices were captured at a midpoint between the outer edge of the acetabulum and the joint surface of the knee.

In this study, 2,724 subjects were analyzed. The SCAPIS subjects included 500 male participants (age $58.3 \pm 4.3$ years, BMI $27.9 \pm 4.2$ kg/$m^2$) and 452 female participants (age $58.5 \pm 4.3$ years, BMI $27.2 \pm 5.4$ kg/$m^2$). The IGT cohort comprised 784 males (age $58.4 \pm 4.5$ years, BMI $28.2 \pm 3.9$ kg/$m^2$) and 988 females (age $57.8 \pm 4.5$ years, BMI $27.4 \pm 4.6$ kg/$m^2$). Age and BMI values are presented as mean $\pm$ standard deviation (std).

### 3.2. Image segmentation

Image segmentation was used for two main purposes in this work. Firstly, segmentation masks corresponding to the liver, VAT, and thigh muscles were generated by CNN-based segmentation models (UNET++) (Ahmad et al., 2023). These masks were used to estimate the area ($cm^2$) of the liver, VAT, and thigh muscle, parameters which were used as prediction targets for this feasibility study. The UNET++ models demonstrated very good performance, with high Dice scores for the masks: liver (0.994), VAT (0.937), and thigh muscle (0.996), indicating their suitability as sources of reliable values for the experiments liver, VAT, and thigh muscle in this work. Secondly, the liver and thigh muscle masks, as well as spleen, SAT, and skeletal muscle masks (having Dice scores of 0.993, 0.990, and 0.988 respectively) were used to facilitate deformable image registration.

In addition to the organ and tissue-specific masks, we also created body masks for the liver, abdomen, and thigh slices to remove CT tables appearing in the FOV by applying a threshold to CT intensities above -190 HU. These body masks were used to remove elements lacking relevance from the images, aiding both registration and deep regression.

Finally, to facilitate the image registration, several thresholding-based masks were generated. We extracted vertebra masks in the liver and abdomen slices, where intensities above 200 HU were thresholded, followed by a morphological opening operation to remove noise. In the abdomen slice, the skeletal muscle mask was utilized to generate an intra-subcutaneous adipose tissue (ISAT) mask. For the thigh slices, masks for cortical bone and lean tissues were generated by thresholding above 200 HU for bone and between -29 HU and 150 HU for lean tissues (Broder, 2011; Mitsiopoulos et al., 1998).

### 3.3. Deep regression and cohort saliency analysis

#### 3.3.1. UNCERTAINTY-AWARE DEEP REGRESSION

For deep regression, the ResNet-50 network, as illustrated in Figure 1(a), was trained for quantification of liver area, VAT area, and thigh muscle area (separately per target). Initially trained on the ImageNet database with 1000 classes, we adopted a transfer learning approach (Kim et al., 2022), by modifying the last top layer to customize the network for our specific tasks. To make the network uncertainty-aware, rather than predicting a single target value (point prediction), the network was configured to produce two output values: the mean and variance (uncertainty estimation) of a Gaussian distribution. Initially, the dimensions of each CT image were $512{\times}512{\times}1$. Deep regression was applied to all three images as a collage. This was done for two reasons: (i) to input all image information to the network, and (ii) to test if the model could learn to use the relevant slice (according to prior knowledge) for each prediction target. To form these collages, three slices of liver, abdomen, and thigh were pre-processed to remove the tables and stacked together as illustrated in Figure 1, resulting in a collage with dimensions of $1004{\times}1004{\times}1$. The stacked images were resized and converted into a three-channel input of $512{\times}512{\times}3$, to make the images compatible in format with the pre-trained network.

The network was trained using 10-fold cross-validation, with slight data augmentation consisting of random shifts (up to 16 pixels both horizontally and vertically). We also performed additional experiments with other augmentation techniques including rotation, shear, re-scaling, and elastic deformation. However, the performance dropped when these augmentation techniques were included in preliminary experiments, so we omitted them from the main experiments of this work.

We used a batch size of 16, a specialized mean-variance training loss function (Lakshminarayanan et al., 2016), as described in more detail in the appendix A, and the Adam optimizer (Kingma and Ba, 2014) with a learning rate of 0.0001. The network was trained for 10,000 iterations, and the learning rate was reduced to 0.00001 after 8,000 iterations. The regression pipeline was implemented using the PyTorch framework (Paszke et al., 2019).

As a pre-study, we assessed several networks including VGG-16 (Karen Simonyan, 2014), DenseNet121 (Huang et al., 2017), EfficientNet (Mingxing and Le., 2019), a Transformer-based network (Vaswani et al., 2017), a graph neural network (Naman and David, 2019), and also ResNet-18/50. ResNet-50 was chosen based on its overall best performance across the three targets.

### 3.3.2. Cohort saliency map analysis

To assess common regions of interest across the entire cohort studies, a cohort saliency map analysis was performed. We applied Grad-CAM to generate a saliency map for each prediction. We then transformed each Grad-CAM saliency map into a common reference space using a deformation field obtained by image registration. These saliency maps were then aggregated by taking the voxel-wise mean saliency, and we refer to these aggregate saliency maps as *cohort saliency maps*. They were visually evaluated to observe the common regions highlighted by the network across the entire cohort. We performed this analysis by computing the Grad-CAM saliency maps at stages 2, 3, and 4 of the network.

The registration pipeline (Ahmad et al., 2024) used to facilitate the cohort saliency analysis involved a one-step process utilizing multiple channel inputs simultaneously, including raw input CT images and their corresponding masks. During the registration, a higher weight was assigned to the CT images compared to the masks. To configure the registration method, we select 8 pyramid levels and regularization weight maps used were 0.05 for air, adipose, and soft tissues, while the bone was assigned a weight of 0.1. Image weights were defined as 0.5 for liver and thigh slices and 0.4 for abdomen slices. For body masks, weights were set to 0.1, 0.15, and 0.2. The registration process used the sum of squared differences as an objective function and a displacement field transformation model (Jönsson et al., 2022). For optimization, a fast graph-cut-based method was used (Ekström et al., 2020).

The reference spaces were chosen to be as representative as possible for each group of slices to be registered. To achieve this an approach using z-scores of explicitly quantified organ and tissue depots were used and different template images were selected for the different slices, for males and females in IGT and SCAPIS resulting in a total of 12 different templates (3 slices $\times$ 2 sexes $\times$ 2 studies). For the liver slice, the sum of the z-scores of liver and spleen areas was minimized. For abdomen slices, the sum of the z-scores of VAT, subcutaneous adipose tissues (SAT), and skeletal muscle areas was minimized. Similarly, for thigh slices, the sum of the z-scores of thigh SAT and muscle areas was minimized.

### 3.4. Evaluation

The network was trained for quantification of the liver area($cm^2$), VAT area($cm^2$), and thigh muscle area($cm^2$) by evaluating target predictions in terms of mean and variance (uncertainty) scores. The efficacy was assessed by measuring the quality of fit using the coefficient of determination ($R^2$), which delineates the proportion of variance accounted for by the network. The mean absolute error (MAE), defines the disparity between actual and predicted values. The experimental results were performed on Intel (R) Xeon(R) W-2133 CPU at 3.60 GHz with 32 CPU RAM and Nvidia RTX Ti with 11GB of GPU RAM.

## 4. Results and discussion

The proposed method demonstrated good results overall, as shown in Table 1, with high $R^2$ and low MAE across targets. Notably, VAT and thigh muscle areas showed high $R^2$ scores and low error rates. The uncertainty estimation, represented as variances, generally indicated confidence in the predictions, except for a high variance in the liver area predictions

Table 1: Evaluation of the proposed deep regression for predicting liver, VAT, and thigh muscle areas ($cm^2$) in the IGT (n = 1772) and SCAPIS (n = 952) cohorts. The table presents the target reference mean ($cm^2$), $R^2$, mean absolute error (MAE), and prediction uncertainty (mean ± std).

| Cohort | Target area | Mean | $R^2$ | MAE | Uncertainty |
|--------|-------------|------|-------|-----|-------------|
| **IGT** | Liver | 176.21 | 0.939 | 5.117 | 3.271 ± 1.807 |
| | VAT | 171.48 | 0.997 | 3.431 | 2.071 ± 0.752 |
| | Thigh muscle | 265.98 | 0.998 | 1.593 | 1.682 ± 0.344 |
| **SCAPIS** | Liver | 162.67 | 0.866 | 7.930 | 2.138 ± 0.818 |
| | VAT | 193.32 | 0.995 | 4.726 | 1.955 ± 0.589 |
| | Thigh muscle | 266.99 | 0.997 | 2.149 | 1.702 ± 0.371 |

for the IGT cohort, possibly due to anatomical variation across subjects. For comparison, the model were also trained with mean squared error (MSE) as the loss function and the results are listed in appendix B (Table 2), where we observed lower performance compared to the uncertainty-aware approach.

A comparison of IGT predictions and actual measurement scatter and Bland-Altman plots are illustrated in appendix C (Figure 3) and for SCAPIS in appendix E (Figure 5). In the scatter plot, the x-axis is labeled actual score ($cm^2$), and the y-axis is labeled predicted score ($cm^2$). The green crosses represent individual data points, and the red line is the line of best fit. The predictions fall close to the regression line, suggesting a strong linear relationship between the actual and predicted scores. In the Bland-Altman plot the green crosses represent individual data points, showing the difference between the predicted and actual scores against their average.

The Grad-CAM cohort saliency map analysis, illustrated in Figure 2 for IGT, for SCAPIS illustrated in appendix D (Figure 4). Additionally, a cohort saliency map was generated using random template selection for the IGT study to analyze the robustness of the cohort saliency map to the selected template. The results were found to be consistent with the chosen z-score based template, as demonstrated in the appendix F (Figure 6), highlighting the same targeted regions such as liver, VAT, and thigh muscle areas in both male and female cohorts, as for the z-score based template. Stage 2 includes the initial convolutional layers, batch normalization, ReLU activations, max pooling, and the first two stages of the ResNet-50. Early stages generally capture low-level features such as edges, colors, and basic textures. The highlighted regions include the liver and thigh muscle areas simultaneously, suggesting the presence of shared feature information at that particular stage. We believe that this might indicate a statistical relationship between these areas, used by the network in the prediction of the liver and thigh muscle areas. To evaluate this we performed Pearson correlation analysis on explicit measurements of liver and thigh muscle areas, with correlations of 0.24/0.21 for males/females in SCAPIS, and 0.30/0.38 for males/females in IGT confirming the plausibility of the relationships observed in the saliency analysis. For the VAT target, Grad-CAM highlights the VAT areas. In Stage 3, the network started to capture more complex and high-level features. These layers can recognize more abstract patterns, which are likely more relevant for identifying target regions.

Stage 4 includes all stages up to but excludes the last two layers (which are average pooling

Figure 2: Grad-CAM cohort saliency maps for the IGT cohort using the ResNet-50 network applied to males (n = 784) and females (n = 988). The highlighted region signifies important areas across the cohort. Warmer colors (red and yellow) indicate higher importance; cooler colors (purple and black) indicate lower importance.

and the final fully connected layer). The features at this stage of the network are even more high-level and abstract than stage 3. These layers capture complex patterns with high relevance for solving the prediction task. The higher representational power at this stage could result in more robust and stronger highlighting of relevant regions, at the cost of further reduced spatial resolution. We notice that the spine in liver slices is light up, which the model is picking up on additional areas beyond the target. It could be due to their complex patterns or liver size varying across subjects that the model has learned to recognize as important. Further, this could be a case of the model being too sensitive to certain features, leading to a broader area of highlight than intended.

A strength of this study was that we could demonstrate the feasibility of the cohort saliency by highlighting target regions of interest in the three slice collages. A limitation of the choice of target measurements is that they were all derived from single-slice measurements, requiring no interaction between the information in the different slices to successfully solve the task. To extend this to more complex multi-slice target measurements is an interesting future research avenue. Another limitation is that, although we present cohort saliency results, deep regression is less explainable compared to models based on explicit image-based measurements. Furthermore, successful deep regression requires extensive data; segmentation approaches may be advantageous for explicit feature measurement.

## 5. Conclusion

In this feasibility study, an uncertainty-aware deep regression pipeline with associated Grad-CAM cohort saliency mapping has been successfully developed and evaluated for body composition measurement predictions from three single-slice CT collages.

## Acknowledgments

This study was funded by Swedish Research Council(2019-04756), Heart and Lung foundation, EXODIAB, and an AIDA-SCAPIS innovation project grant.

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

## Appendix A. Loss function for training deep regression with uncertainty estimation

To train the uncertainty-aware deep regression ResNet-50 networks, we used a loss function that takes predictions of mean and variance values for each target as follows:

$$-\log p_\theta(y_n|x_n) = \frac{\log \sigma_\theta^2(x)}{2} + \frac{(y - \mu_\theta(x))^2}{2\sigma_\theta^2(x)} + \text{constant}. \tag{1}$$

In Equation (1), $\mu_\theta(x)$ and $\sigma_\theta^2(x)$ denote the model's predicted mean and variance, respectively, with $\theta$ representing the set of parameters. This form of the loss function is adapted from (Lakshminarayanan et al., 2016), and it is commonly used in regression models for parameter estimation by minimizing the negative log-likelihood, under the assumption of Gaussian-distributed errors.

## Appendix B. ResNet-50 model with MSE as loss function

Table 2: Evaluation Metrics of proposed (ResNET-50) model using MSE loss function for target prediction of liver, abdomen VAT, and thigh muscle areas ($cm^2$), the IGT (n = 1772) and SCAPIS (n = 952) cohorts. The table presents the target reference mean ($cm^2$), $R^2$, and mean absolute error (MAE).

| Cohort | Target | Mean | $R^2$ | MAE |
|--------|--------|------|-------|-----|
| **IGT** | Liver | 176.21 | 0.932 | 5.572 |
| | VAT | 171.48 | 0.996 | 3.714 |
| | Thigh muscle | 265.98 | 0.998 | 1.727 |
| **SCAPIS** | Liver | 162.67 | 0.859 | 8.245 |
| | VAT | 193.32 | 0.994 | 5.579 |
| | Thigh muscle | 266.99 | 0.996 | 2.457 |

To evaluate if the uncertainty-aware approach has an advantage for the prediction performance, the network was also trained with mean squared error (MSE) as the loss function and the results are shown in Table 2. The networks trained with the mean-variance loss function have slightly better performance in terms of of MAE error and $R^2$ results.

## Appendix C. Scatter and Bland Altman plots for IGT cohort

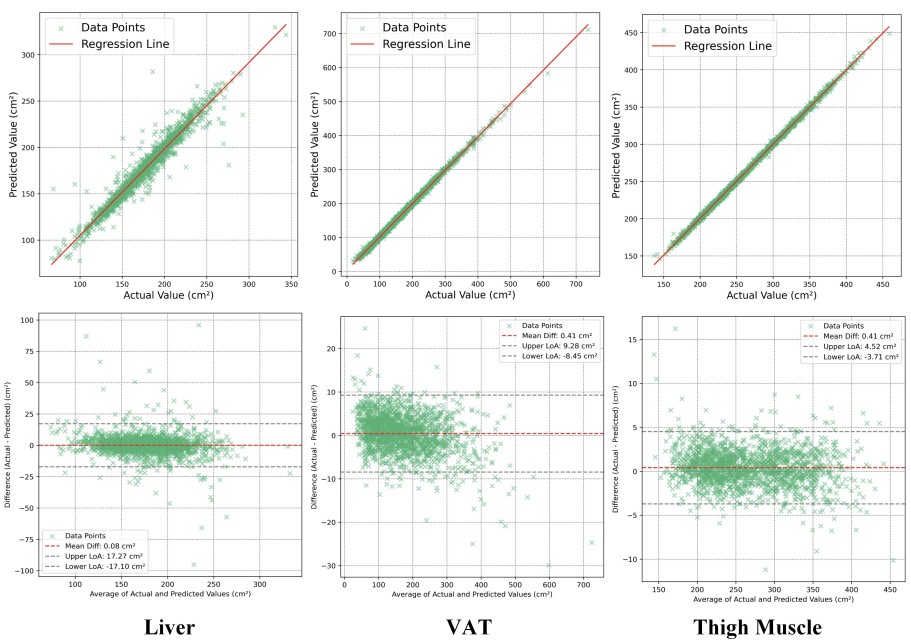

Figure 3: Comparative analysis of predicted and actual measurement scores in $cm^2$: The scatter plot with a regression line and Bland-Altman plot (Agreement assessment with 95% confidence) for IGT collages (n = 1772).

## Appendix D. Grad-CAM cohort saliency maps for SCAPIS cohort

The Grad-CAM cohort saliency for the SCAPIS cohort, the targeted measurements liver, VAT, and thigh muscle areas at stages 2, 3, and 4 of the ResNet-50 model is illustrated in Figure 4.

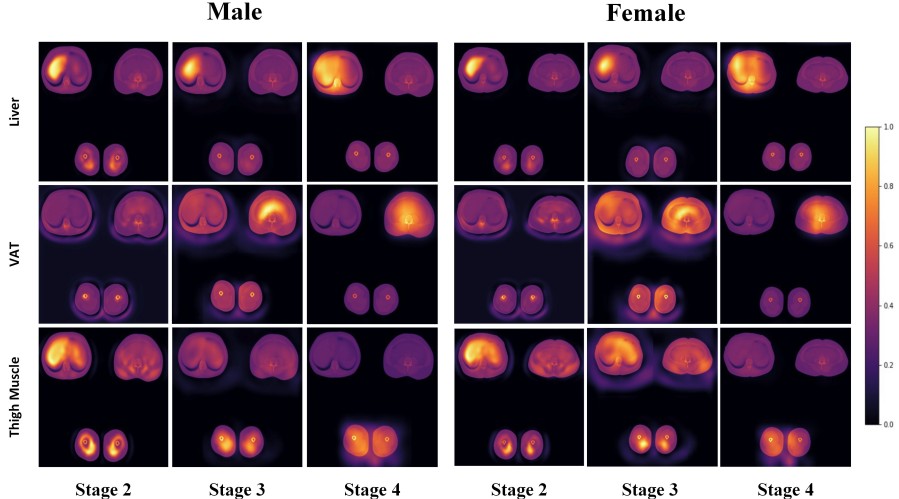

Figure 4: Grad-CAM cohort saliency maps of the SCAPIS cohort at stages 2, 3, and 4 of the ResNet-50 network applied to collages corresponding to males (n = 500) and females (n = 452). The highlighted region signifies important areas across the cohort.

## Appendix E.  Scatter with regression line and Bland-Altman plots for SCAPIS cohort

A comparison of SCAPIS predictions and actual measurements are illustrated in Figure 5. In the plot, the x-axis is labeled actual score $(cm^2)$, and the y-axis is labeled predicted score $(cm^2)$. The green crosses represent individual data points, and the red line is the line of best fit. The predictions fall close to the regression line, suggesting a strong linear relationship between the actual and predicted scores. Bland-Altman plot the green crosses represent individual data points, showing the difference between the predicted and actual scores against their average. The red dashed line represents the mean difference, which is very close to 0 $cm^2$, indicating good agreement. The black dashed lines are the upper and lower limits of agreement (LoA), which show the range in which most differences lie.

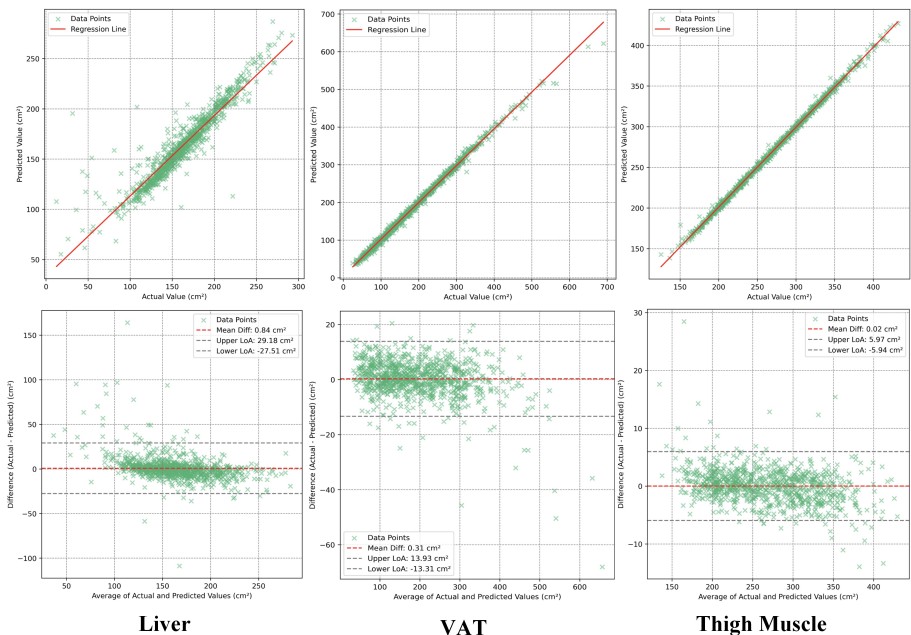

Figure 5: SCAPIS collages (n = 952), Comparative analysis of predicted and actual measurement scores in $cm^2$: The scatter plot with a regression line and Bland-Altman plots (Agreement assessment with 95% confidence).

# Appendix F. Randomly Selected Template Based Cohort Saliency Map

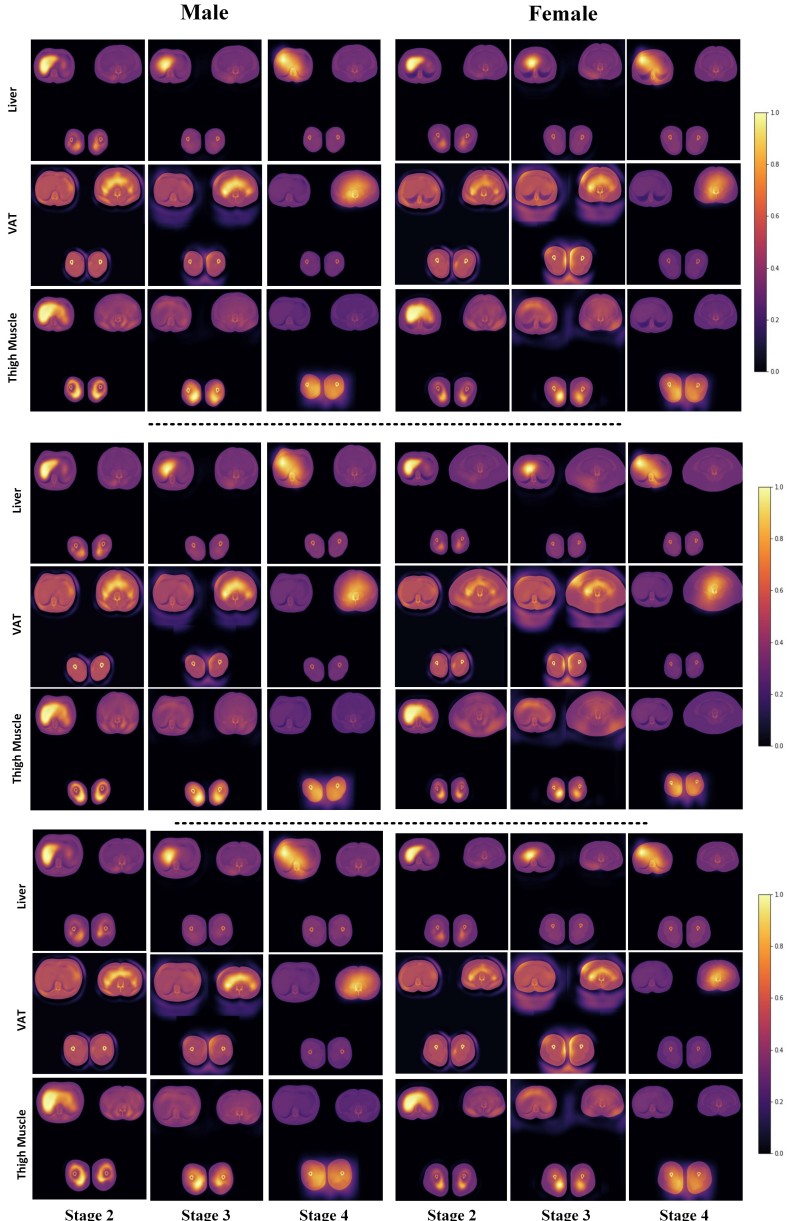

Figure 6: Randomly selected template(per slice) based Grad-CAM cohort saliency maps for the IGT cohort, males (n = 784) and females (n = 988). The highlighted region signifies important areas across the cohort. A dashed line separates the set of collages.

