# OpenReview forum: "Interpretable Uncertainty-Aware Deep Regression with Cohort Saliency Analysis for Three-Slice CT Imaging Studies"
_MIDL.io/2024/Conference — MIDL 2024 Poster_

### Official Review · Reviewer_nXzM · 2024-02-28

**Confidence:** 3
**Preliminary Rating:** 2
**Recommendation:** Poster
**Final Rating:** 2.5

**Summary:**

The paper train and evaluate an uncertainty-aware deep regression based on ResNet-50 for quantification of cross sectional areas of liver, visceral adipose tissue (VAT), and thigh muscle, with outputs its prediction mean and variance. The model was evaluated using R2 and MAE as metrics for predictions, showing high R2 and low MAE across targets. Additionally, the model was evaluated with saliency analysis based on Grad-CAM. Deformable image registration to a template subject was used to get group-wise cohort saliency analysis.

**Strengths:**

The paper is clearly written. The author used the well-known ResNet-50 for quantification of liver area, abdomen VAT area, and thigh muscle area. The training procedure was clearly described. To better interpret the network, the author used Grad-CAM saliency map for model interpretability. In addition, the paper has enough statistic analysis for model performance evaluation.

**Weaknesses:**

The paper doesn't show much novelty. The paper seems just apply ResNet-50 and Grad-CAM saliency map to another application, to   quantify the liver area, abdomen VAT area. In addition, the paper fail to do comprehensive experiments such as why trainning one for all rather than 3 networks etc.

**Detailed Comments:**

The paper trained one network for liver area, VAT area and thigh muscle area quantification. I would like to suggest to compare with 3 networks respectively for liver area, VAT area and thigh muscle area quantification. From Fig3 Grad-CAM cohort saliency maps, some cases of saliency maps in the thigh muscle showed highlighted region in the liver, which might shows the once-for-all network might not be enough to capture the inherent information. The network used two datasets, SCAPIS and IGT. They two datasets might have distribution difference. Therefore, the author should provide data pre-processing to harmonize the two datasets. If the two datasets have dramatic difference, it is better to conduct some experiments, such as training separately, training all data etc. The authors mentioned that the network was trained with slight data augmentation consisting of random shifts, I would like to suggest to try some other data augmentation as well to make the network robust for perturbations.

**Justification Of Final Rating:**

I want to express my appreciation for addressing my comments as well as the suggestions from the other reviewer. The revised manuscript  incorporated additional experiments which enhanced the clarity of important points. But still I feel the paper lacks originality and novelty.

**Justification Of The Preliminary Rating:**

Based on my best, I acknowledged that the authors proposed a deep learning methods for liver area, VAT area and thigh muscle area quantification. However, the paper lacks novelty since the paper seems just apply ResNet-50 and Grad-CAM saliency map to a specifical application.

**Questions To Address In The Rebuttal:**

Could you explain why you trained one network for you? I would like to the authors to perform the comparison of once-for-all network with 3 networks respectively for liver area, VAT area and thigh muscle area quantification. The comparison includes the R2, MSE, and saliency maps. In addition, train another network only to output mean, and do comparison. Last, try some other data augmentation and see whether it benefit the network performance.

**Special Issue:**

No

---

> ### Author Response · Authors · 2024-03-17
>
> Question: Could you explain why you trained one network for you? I would like to the authors to perform the comparison of once-for-all network with 3 networks, respectively, for liver area, VAT area, and thigh muscle area quantification.
>
> Reply: Thank you for raising this point. The training of a model for a single individual target is likely to enhance the model performance, by avoiding interference between the features learned to solve each individual task. Though sometimes multi-task learning can be beneficial, the targets of interest here are derived from different slices, and therefore unlikely to benefit from shared features. Conversely, training the model with multiple targets tends to increase network complexity and may reduce performance. Also training each model individually allows for easier evaluation and interpretation of results. We can assess each model's performance independently, providing clearer insights into how well each target is being addressed, and what features contribute to the prediction.
>
> Furthermore, as we mentioned in the manuscript, this is a proof-of-concept study that allows us to highlight relevant target regions in the image with precision while interpreting the output, as we know what to expect/where to look. All in all, training a multi-target network for this purpose could introduce noise for each individual target prediction, and potentially decrease the performance.
>
>
> Question: In addition, train another network only to output mean, and do comparison.
>
> Reply: Thank you for your feedback. We have now incorporated results from models trained with MSE into the appendix, see Appendix B.  The results from the proposed model utilizing the mean-variance training loss function show slightly improved MAE error and R2 outcomes compared to those obtained with MSE, thus providing both uncertainty estimates that we can use to assess how robust our predictions are, as well as obtain higher prediction performance.
>
> Question: Last, try some other data augmentation and see whether it benefit the network performance.
>
> Reply: For data augmentation, we have experimented with various techniques including rotation, shear, scaling, and elastic deformation. However, the network's performance did not improve from inclusion of these data augmentations, but the performance decreased. A statement about this observation has been added to the manuscript (see section 3.3.1).

---

### Official Review · Reviewer_5KS6 · 2024-02-28

**Confidence:** 4
**Preliminary Rating:** 4
**Recommendation:** Poster
**Final Rating:** 5

**Summary:**

This is a feasibility study that uses CT imaging to understand body composition and its related factors. The study utilizes deep learning techniques to segment and measure different types of tissues and organs, and to aid in the estimation and prediction of body composition parameters. The study also uses saliency maps to highlight areas of high importance to the prediction targets, which reinforces trust in the methodology and the proposed deep regression method. The study provides group-wise visualization of image regions of importance for associations to biomarkers of interest. The study is important for understanding metabolic health and preventing chronic diseases.

**Strengths:**

First, this paper proposes a new method to analyze body composition using CT images. This method combines deep regression and uncertainty estimation to produce more accurate results. Second, this paper uses cohort saliency analysis to visualize important regions in the images, which allows us to determine their relationship with body composition and metabolic health. This method helps in interpreting image analysis results. Third, this paper uses deep learning techniques to effectively segment and measure various tissues and organs, such as liver, abdominal fat, thigh muscles, etc.

**Weaknesses:**

One of the main weaknesses of the paper is that it lacks a comparison with existing state-of-the-art methods. While the paper does compare its proposed method with a few baseline methods, it does not compare its results with other recently published methods in the field. This limits the ability to assess the true novelty and potential impact of the proposed method.

Another weakness of the paper is that it does not provide a detailed explanation of the limitations of the proposed method. Why did the authors use Deep Regression with ResNet-50 instead of another models--the inclusion of such defense will improve this paper.

Furthermore, the paper could benefit from a more detailed explanation of the methodology used for cohort saliency analysis. For example, how the saliency maps were generated and how they were used to identify regions of interest will reinforce this paper.

Also, Conclusion section of the paper seems insufficient.

**Detailed Comments:**

1. The paper could benefit from a detailed explanation of the usage of the proposed method.
2. The paper could benefit from a more detailed explanation of the limitations of the proposed method.
3. The paper could also include a more thorough comparison with existing state-of-the-art methods.

**Justification Of Final Rating:**

This is a feasibility study that uses CT imaging to understand body composition and its related factors. The study utilizes deep learning techniques to segment and measure different types of tissues and organs, and to aid in the estimation and prediction of body composition parameters. The study also uses saliency maps to highlight areas of high importance to the prediction targets, which reinforces trust in the methodology and the proposed deep regression method. The study provides group-wise visualization of image regions of importance for associations to biomarkers of interest. The study is important for understanding metabolic health and preventing chronic diseases.

The authors gave answers to my comments in weakness quite properly, especially in the comparison of other methods even if it is only descirption instead of a quantitative one. considering the potential outcome in the future, I will raise my evaluation for this paper from the inital one ( 4: Weak accept) by one step higher.

**Justification Of The Preliminary Rating:**

Clinically this paper is quite interesting while the methodology and its justification for the adoption is of question. Due to clinical novely, I give 4 out of 5 with a recommendation to a poster section.

**Questions To Address In The Rebuttal:**

The lack in comparison study might be due to their originality and novelty. However, the authors might consider a similar study and their methods and results for the comparison purpose.

**Special Issue:**

No

---

> ### Author Response · Authors · 2024-03-17
>
> Question: The paper could benefit from a detailed explanation of the usage of the proposed method.
>
> Reply: The proposed method is specifically tailored for SCAPIS and IGT, employing three single-slice images to establish correlations between imaging and non-imaging data. By training a regression model, it predicts the liver area, fat, and muscle composition of a subject, utilizing these predicted measurements for further analysis in the cohort study with follow-up assessments.
>
> Our pipeline is a proof of concept for the liver area, VAT, and thigh muscle. The cohort saliency analysis based on Grad-CAM has shown that our model accurately identifies these important regions during cohort saliency analyses, paving the way for integrating imaging data with other biomarkers.
>
> In the future, we plan to use this approach to study, in detail, deep-learning based links and associations between imaging data and disease-related biomarkers of interest, for example, related to diabetes, age, and cardiovascular disease.
>
> Question: The paper could benefit from a more detailed explanation of the limitations of the proposed method.
>
> Reply: While we do present cohort saliency results, it's important to note that our approach may be less explainable compared to models relying on explicit image-based measurements. Additionally, the reliance on deep regression requires extensive data. Although segmentation approaches offer advantages for explicit feature measurement, it's worth acknowledging that our current methodology serves as a proof of concept, as mentioned in the paper. In future work, we aim to incorporate additional parameters such as BMI and age, which may not be feasible with segmentation techniques.
>
> Question: The paper could also include a more thorough comparison with existing state-of-the-art methods.
>
> Reply: In our paper, we highlighted that it is a proof of concept tailored for SCAPIS and IGT studies that use three-slice CT images. Our goal was to create a pipeline for these specific cohorts. Early on, we assessed several networks, including VGG-16, EfficientNet, DenseNet121, transformer based model, graph neural networks, and ResNet-18. ResNet-50 was chosen as our baseline for its optimal performance for all targets. In future work, we aim to enhance the pipeline by incorporating more non-imaging parameters and biomarkers, as ResNet-50 shows the best overall performance for various targets. A statement about this observation has been added to the manuscript (in Section 3.3.1).

---

> > ### Comment · Reviewer_5KS6 · 2024-03-27
> > **raise my evaluation from the initial one**
> >
> > The authors gave answers to my comments in weakness quite properly, especially in the comparison of other methods even if it is only descirption instead of a quantitative one. considering the potential outcome in the future, I will raise my evaluation for this paper from the inital one ( 4: Weak accept) by one step higher.

---

### Official Review · Reviewer_ifvb · 2024-02-29

**Confidence:** 4
**Preliminary Rating:** 4
**Recommendation:** Poster
**Final Rating:** 4

**Summary:**

The authors present a study where the body composition (measuring thigh muscle, liver, and VAT areas) is inferred using three CT slices at pre-specified locations jointly fed to a ResNet50 network. The model includes an additional output to the regression head, giving an uncertainty estimate assuming a Gaussian error distribution. The study also includes a population-based saliency map analysis for which Grad-CAM outputs were registered to a common space.
Cross-validation was used to evaluate the method on two datasets (SCAPIS and IGT).

**Strengths:**

- Clear experimental setup
- Figures are prepared well and contribute to the understanding of the study/text
- Large sample size with two included datasets
- Population-based analysis and clear evaluation metrics
- Experimental setup is described in detail and easy to follow

**Weaknesses:**

- I would appreciate it if you could compare your approach and results to the previous works you mention and cite in the section on related works. For someone not too familiar with this specific task, it would be great to be able to gauge the performance you report
- The uncertainty estimates are only discussed very briefly. It would be great if you could add some information if, e.g., the uncertainties match the deviations you got measured by the MAE.
- In my opinion, the title is a bit over-promising regarding the interpretability. Judging from the title, I was expecting a non-standard interpretable approach apart from using Grad-CAM.

**Detailed Comments:**

-

**Justification Of Final Rating:**

Thank you for the replies to my comments and the other reviewer's suggestions, and the effort you put into revising the manuscript and adding additional experiments. I believe it has improved and important points have been made clearer.

**Justification Of The Preliminary Rating:**

The study design seems solid, but I would appreciate more information on how the results were achieved compared to the SOTA or clinical relevance. The experimental setup is sound, but only includes a single experiment without further comparisons to competing methods or approaches.

**Questions To Address In The Rebuttal:**

- Can you comment on the clinical impact of the accuracy/error you are making with your approach? I.e., how does this compare to what is useful in practice and what others have achieved?
- Did you consider looking into Bland-Altman plots to see if there is a constant offset in your regression results?

**Special Issue:**

No

---

> ### Author Response · Authors · 2024-03-17
>
> Question: Can you comment on the clinical impact of the accuracy/error you are making with your approach? I.e., how does this compare to what is useful in practice and what others have achieved?
>
> Reply: Thank you for your question regarding the clinical impact of our model's accuracy. It is important to note that our study is primarily contributing to epidemiological research rather than clinical application. We have conducted a proof of concept study of a method for body composition analysis using three-slice CT images.
>
> Our model's accuracy in terms of metrics such as R-squared and mean absolute error has shown promising results using 3 slices of imaging data for epidemiological research, as it can provide a solid foundation for understanding body composition changes over time for SCAPIS and IGT. While the direct clinical impact is not the goal, the precision of our model is important for accurate longitudinal follow-up measurements.
>
> The relatively good performance suggests that we will be able to use the developed technology in further research related to predictions related to research on type 2 diabetes, cardiovascular disease, and associated risk factors. At this high level of accuracy/precision, the variance induced by imaging (single slice only and variance in positioning) would also contribute a significant proportion of the total method variance.
>
> Question: What others have achieved?
> Reply: Our study demonstrates the quantification of visceral adipose tissue (VAT) and thigh muscle area using 3 single-slice CT images. Specifically, we achieved an R² of 0.997 for VAT and 0.998 for thigh muscle area in the SCAPIS cohort, with mean absolute errors (MAE) of 3.431 cm² for VAT and 1.593 cm² for thigh muscle area. In contrast, the author in paper [1], in their supplementary material, reported R² values of 0.994 for VAT and 0.993 for thigh muscle, with MAEs of 0.122 L and 0.162 L, respectively, using deep regression with ResNet50 on UK Biobank data.
> It's important to note the different units of measurement (cm² in our study vs. litres in the referenced study) and the imaging modality used (CT vs. MRI). Despite these differences, both studies underscore the high accuracy and reliability of deep learning models in body composition analysis. Our results further build upon these findings by demonstrating the utility of CT imaging limited to 3 single slices only and highlighting the model's precision, as evidenced by lower MAEs in predicting VAT and thigh muscle areas, showing the effectiveness of our methodology in body composition.
>
> Reference:
> [1]. Langner, Taro, et al. "Uncertainty-aware body composition analysis with deep regression ensembles on UK Biobank MRI." Computerized Medical Imaging and Graphics 93 (2021): 101994. DOI. https://doi.org/10.1016/j.compmedimag.2021.101994
>
> Question: Did you consider looking into Bland-Altman plots to see if there is a constant offset in your regression results?
>
> Reply: Thank you for your valuable suggestion to incorporate Bland-Altman plots to assess the presence of a constant offset in our regression results. Following your advice, we have now included Bland-Altman plots in Appendix C and Appendix E, given the page limit constraint for the main manuscript.

---

### Official Review · Reviewer_PxQ4 · 2024-03-02

**Confidence:** 5
**Preliminary Rating:** 1
**Recommendation:** Poster

**Summary:**

A 2D slice-wise CT based analysis of liver, abdomen and thigh regions to estimate the adiposity of the patients to provide quantification of metabolic health of patients is performed using a relatively large dataset 2,724 patients. A 2D Resnet classifier was created to predict the mean and standard deviation of the metabolic volume to provide a measure of uncertainty in the estimate of the adiposity when using 3 specific slices corresponding to the iver, abdomen, and the thighs. GraCam approach was combined with a cohort saliency analysis to show the which regions were implicated in making the prediction.

The methodology is incremental. Key details of the analysis including how the resnet was trained to predict mean and standard deviation to model uncertainty, how were the individual 3 slices selected, registration method, selection of the canonical reference etc are all missing. There may be some conceptual novelty regarding the automated estimation of the metabolic health.

**Strengths:**

* There is some conceptual novelty regarding the automated estimation of the metabolic health. Automated and objective estimation of adiposity in patients can have clinical implications in managing patients.
* A relatively large dataset of 2,724 patients with similar number of males and females were used in teh analysis.

**Weaknesses:**

* The methodology is incremental.
* Key details of the analysis including how the resnet was trained to predict mean and standard deviation to model uncertainty, how were the individual 3 slices selected, registration method, selection of the canonical reference etc are all missing.
* How robust is the method to the selection of the individual slices?
* How robust is it to the selection of the canonical patient reference.
* How was the registration performed between cohorts if discrete sets of slices from each patient was used in the analysis.

**Detailed Comments:**

Please see my comments in the summary and weaknesses.

**Justification Of The Preliminary Rating:**

* The methodology is incremental.
* Key details of the analysis including how the resnet was trained to predict mean and standard deviation to model uncertainty, how were the individual 3 slices selected, registration method, selection of the canonical reference etc are all missing.
* How robust is the method to the selection of the individual slices?
* How robust is it to the selection of the canonical patient reference.
* How was the registration performed between cohorts if discrete sets of slices from each patient was used in the analysis.

**Questions To Address In The Rebuttal:**

* The methodology is incremental. What is the novelty here?
* Key details of the analysis including how the resnet was trained to predict mean and standard deviation to model uncertainty, how were the individual 3 slices selected, registration method, selection of the canonical reference etc are all missing.
* How robust is the method to the selection of the individual slices?
* How robust is it to the selection of the canonical patient reference.
* How was the registration performed between cohorts if discrete sets of slices from each patient was used in the analysis.

**Special Issue:**

No

---

> ### Author Response · Authors · 2024-03-17
>
> Question: The methodology is incremental. What is the novelty here?
>
> Reply: Thank you for your query regarding the novelty of our methodology. Our approach is distinctive in that it integrates an uncertainty-aware deep regression model using ResNet-50 with cohort saliency mapping via Grad-CAM, specifically for body composition analysis with three-slice CT images—a method not previously explored in metabolic health studies with multiple single slices.
>
> There is also novelty in the image registration and in the evaluation of the cohort saliency maps at different stages of the ResNet-50 network. Details on critical components of the image registration have now been added to this paper. (The complete details of the image registration and its evaluation are in a paper under review.)
>
>
> Question: Key details of the analysis including how the resnet was trained to predict mean and standard deviation to model uncertainty, how were the individual 3 slices selected, registration method, selection of the canonical reference etc are all missing.
>
> Reply: We appreciate the reviewer's request for further methodological details. Our uncertainty-aware deep regression model was trained using a specialized loss function that allows the network to predict both the mean (predicted score) and variance (uncertainty in prediction) for each target parameter. This loss function, adapted from Lakshminarayanan et al. (2016), is particularly effective for regression models that require the estimation of parameters under the assumption of Gaussian-distributed errors. For more details, please refer to Appendix A, where we describe the loss function.
>
> Regarding the three CT slices, we realize that this might have been a source of misunderstanding. The three CT slices collected are determined during the imaging sessions according to the SCAPIS and IGT study imaging protocols. We now have clarified this more in Figure 1 and its caption as well as in the introduction the first time the three slices are mentioned.
>
> The registration method is now described in more detail. For a more in-depth explanation of the selection of the reference images please see sections 3.2 and 3.3.
>
>
> Question: How robust is the method to the selection of the individual slices?
>
> Reply: As previously clarified, the methodology described in this paper does not involve specific methods for selecting individual slices. Instead, the study is based on a dataset consisting of three predetermined slices from the SCAPIS and IGT cohorts, with standardized positions in the liver, abdomen, and thighs. As we do not have access to volumetric data in these studies we can unfortunately not analyze the precision and robustness to change in slice positioning.
>
>  Question: How robust is it to the selection of the canonical patient reference.
>
> Reply: We thank the reviewer for this very good suggestion. We have now redone the cohort saliency analysis three times per sex with randomly selected template slices. The results from these new experiments highlighted similar regions for all three targets as our proposed z-score-based approach. The results are added in Appendix Section F.
>
> Question: How was the registration performed between cohorts if discrete sets of slices from each patient was used in the analysis.
>
> Reply: Inter subject registrations are always challenging. The standardized positioning of the imaged slices was indeed critical for the standardization of the image content. The registrations in this work are always done within cohorts and within the same sex.

---

### Comment · Area_Chair_Ehz3 · 2024-03-15
**Gentle reminder of rebuttal deadline**

Dear authors,

This is just a gentle reminder that the rebuttal deadline is coming up. I encourage you to take this opportunity so you can engage with the reviewers in the upcoming discussion phase.

Best wishes,
AC

---

### Meta-Review · Area_Chair_Ehz3 · 2024-04-03

**Recommendation:** Accept (Poster)
**Confidence:** 2

**Metareview:**

This paper proposes a framework to predict the area of different image regions (e.g. of the liver) from CT slices using deep regression. The methodology is a fairly straightforward application of existing approaches, but the work is clinically well motivated and implemented carefully.

Overall, the paper received very mixed reviews. Unfortunately, the most critical reviewer did not respond to the rebuttal, but I had a careful look at the paper, review, and rebuttal myself. I believe the authors addressed the raised concerns adequately. Two reviewers were concerned about the limited methodological novelty. As this is more of a translational paper, I would tend to disregard this, although I agree the technical contributions are limited.

While not groundbreaking, I believe this is a sound application paper with an interesting and clinically relevant application. As in my opinion, the authors responded well to the negative reviews, I tend to follow the positive reviewers and can recommend acceptance of the paper.

---

### Decision · Program_Chairs · 2024-04-06

Accept (Poster)